# Touching a NeRF: Leveraging Neural Radiance Fields for Tactile Sensory Data Generation

**Shaohong Zhong, Alessandro Albini, Oiwi Parker Jones, Perla Maiolino, Ingmar Posner**
Oxford Robotics Institute
University of Oxford, United Kingdom
{shaohong,alessandro,oiwi,perla,ingmar}@robots.ox.ac.uk

**Abstract:** Tactile perception is key for robotics applications such as manipulation. However, tactile data collection is time-consuming, especially when compared to vision. This limits the use of the tactile modality in machine learning solutions in robotics. In this paper, we propose a generative model to simulate realistic tactile sensory data for use in downstream tasks. Starting with easily-obtained camera images, we train Neural Radiance Fields (NeRF) for objects of interest. We then use NeRF-rendered RGB-D images as inputs to a conditional Generative Adversarial Network model (cGAN) to generate tactile images from desired orientations. We evaluate the generated data quantitatively using the Structural Similarity Index and Mean Squared Error metrics, and also using a tactile classification task both in simulation and in the real world. Results show that by augmenting a manually collected dataset, the generated data is able to increase classification accuracy by around 10%. In addition, we demonstrate that our model is able to transfer from one tactile sensor to another with a small fine-tuning dataset.

**Keywords:** Camera-based tactile sensing, cross-modal tactile data generation

## 1 Introduction

Humans rely heavily on tactile sensing for tasks such as identifying and grasping objects (e.g. picking keys from a pocket) [1, 2]. In this context, tactile sensing is fundamental to retrieve contact information or properties of the object such as roughness or stiffness, and is also able to complement vision in occluded scenarios [1]. Tactile sensing is also critical for robotics applications such as manipulation and control [3] and object or texture recognition [4]. These tasks, especially those related to tactile-based object recognition, are usually tackled with machine learning methods that typically require large amounts of data for training [1, 5, 6]. However, collecting tactile data is challenging as the robot needs to physically interact with the environment and the object. While cameras can capture the global shape of an object, tactile sensors can only capture local features, and a long and time-consuming exploration procedure is usually required to capture the whole shape [7]. Beyond the problem relating to tactile exploration, tactile sensing is also still lacking standards at the hardware level [8]. For the same physical stimulus, the output of different tactile systems can differ significantly, thus limiting the validity of the collected data to a specific sensing technology.

Given the difficulties of tactile data collection, the problem of generating synthetic tactile sensor responses from data acquired using different modalities (which are easier to collect) becomes relevant. In particular, recent works show that vision data (RGB-D images) contain rich sensory information that can be used to generate tactile data [9, 10, 11]. Given a camera or depth image of the object surface as input, these approaches can generate the corresponding tactile sensor output. One limitation of these vision-based generative approaches is that they require the collection of visual samples at given positions and orientations to generate the corresponding synthetic tactile data [9]. However, with the development of neural volume-rendering techniques such as NeRF [12], it is now possible to synthesise high-quality RGB-D images for novel view orientations of a scene, given only sample 2D images and their associated camera poses [12]. In this way, NeRF provides additional information on the structure of the scene that we leverage for generating tactile images for 3D objects.

6th Conference on Robot Learning (CoRL 2022), Auckland, New Zealand.

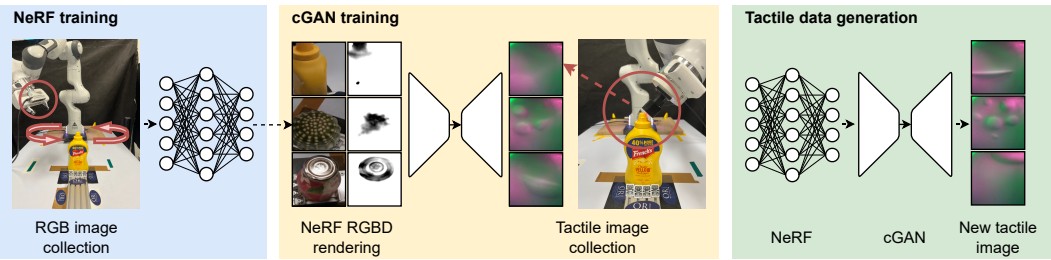

Figure 1: Overview of the framework. We first capture camera images around the object to train a NeRF model. We then collect a tactile image dataset to train a conditional GAN model that takes NeRF-rendered RGB-D images as input and outputs tactile images. Once the cGAN model is trained, we can query NeRF models with arbitrary poses, and pass the RGB-D output to the cGAN model to obtain the desired tactile image from the same orientation as the query pose.

The main contribution of this paper is to introduce a novel framework for leveraging the RGB-D images that NeRFs render to generate tactile sensory data via a deep generative model. As far as the authors are aware, ours is the first work that learns a generative model conditioned on RGB-D inputs for the generation of tactile data. Compared to simulation-based approaches for tactile data generation, our framework removes the need for accurate hand-engineered modelling and calibration of the tactile sensor as well as the objects of interest. Compared to other learning-based approaches, our method is capable of handling 3D objects of different geometries, and is able to generalise to novel views of the object through the use of NeRF models. The use of NeRF models also removes the need for an RGB-D camera to manually obtain a new input image each time a new object reading is desired. In addition, our framework is capable of transferring to a new sensor with a small fine-tuning dataset through the use of a deep generative model conditioned on a sensor background image. In this way, we can overcome the domain gap between different tactile sensors, and open up the potential to leverage different tactile datasets for even better performance.

## 2 Literature Review

This section reviews the approaches that simulate or generate data for camera-based tactile sensors [13, 14]. These devices use a camera to capture the deformation of a soft medium and output a *tactile image*. One class of approaches are based on the development of accurate simulators for camera-based tactile sensors, which aim to make the robot learn tactile features in a simulated environment and transfer the knowledge to the real world [15, 16, 17, 18]. For example, the TACTO simulator presented in [15] allows one to emulate both Digit and the OmniTact sensors by simulating contacts in an off-the-shelf physics engine [14, 19]. Different simulators, such as Taxim aim to simulate the responses of a GelSight sensor by using a polynomial look-up table [16]. Another approach is to leverage methods such as physics-based rendering [17] and depth-maps from physics simulators [18]. Additional works attempt to address the sim2real issue in these tactile simulators through the use of CyclyGANs [20] and texture generation networks [21]. Even though tactile simulators present an attractive option for data synthesis, these tactile simulators require accurate modelling and calibration of the specific tactile sensor via additional calibration equipment [15, 16]. An accurate object model is also needed for simulating contacts [16]. In contrast, our proposed learning-based framework removes the need for modelling the sensor or the object.

A separate class of work explores learning-based approaches. A majority addresses the problem of generating tactile data from vision, using camera-based sensors, as both modalities encode information using the same data structure [10, 22, 9, 11]. For example, Li et al. [10] estimate the response of a GelSight sensor using a cGAN model [23] conditioned on a vision sequence that captures the robot touching the object. However, their method requires a robot to perform the touching action to generate the tactile response. Patel et al. [22] leverage depth sensors and the object mesh to generate tactile images, similarly building on a cGAN model. Gao et al. [24, 25] also attempt to render tactile images of individual objects directly using a NeRF-like model. However, their approach is incapable of generalising to new objects because a new NeRF model needs to be trained on tactile images for the new object. Lee et al. [9] employ a cGAN to generate tactile images from vision, and vice-versa, on a dataset containing top-down views of flat clothes – our work is most similar in spirit

to this. However, we differ significantly in our use of RGB-D images rendered from NeRF models. This removes the need to retake a camera image for each desired tactile image. Our problem setting is also harder, as it focuses on 3D objects with complex geometries. In addition, we demonstrate the transfer of the trained cGAN model to generate data for a new tactile sensor, which, as far as the authors are aware, is the first that demonstrates such generalisation capability across tactile sensors.

## 3   Methods

As seen from Figure 1, our approach involves first training individual NeRF models for objects of interest. Although they only require RGB images for training, the use of NeRFs enables us to learn the 3D structure of the scene and render RGB-D images from arbitrary viewpoints. Then, we propose to train a cGAN model to generate tactile data conditioned on both the RGB-D images rendered by NeRF models and a reference background image for the tactile sensor. To generate tactile data from novel poses, we simply render corresponding RGB-D images from an object's NeRF model and pass the results as inputs to the cGAN model.

### 3.1   Neural radiance fields

A Neural Radiance Field (NeRF) is a model for synthesising novel views of scenes [12]. It requires a set of RGB images and their associated camera poses for training. A trained NeRF is able to synthesise high-quality and novel views of the modelled scene. Formally, a NeRF model represents a continuous scene as a function $F$, which takes the 3D location of a point $\mathbf{m} = (x, y, z)$ and a 2D viewing direction $\mathbf{d} = (\theta, \phi)$ as input, and predicts the RGB colour $\mathbf{c} = (r, g, b)$ and volume density $\sigma$ of the point from the particular viewing direction. The learned function is parameterised using a multi-layered perceptron $F_\Theta$. Using $F_\Theta$, images of the scene can be rendered from arbitrary camera poses. Given the camera center $\mathbf{o}$, the color of each pixel is estimated by summing $N$ samples along the camera ray $C(\mathbf{r})$, where $C(\mathbf{r}) = \mathbf{o} + t\mathbf{d}$, using a numerical quadrature approximation [26]:

$$\hat{C}(\mathbf{r}) = \sum_{n=1}^{N} T_n (1 - \exp(-\sigma_n \delta_n)) \mathbf{c}_n \tag{1}$$

where $T_n = \exp(-\sum_{n'=1}^{n-1} \sigma_{n'} \delta_{n'})$ and $\delta_n = t_{n+1} - t_n$. By predicting the volume density of sampled points in this process, a NeRF can trivially render the depth image of the scene [12], a capacity that we take advantage of in this study. After rendering the required RGB-D image, we perform simple preprocessing by centre cropping of the RGB-D image and normalisation of the depth image before passing the RGB-D image as input to the cGAN model.

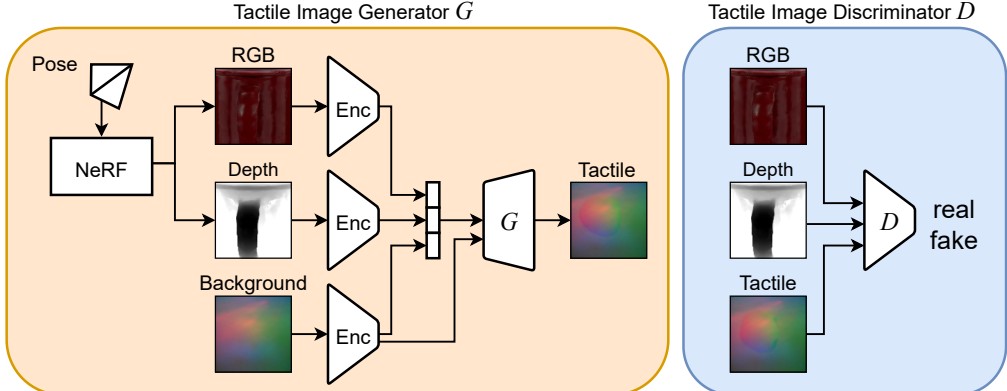

Figure 2: Proposed cGAN architecture. Given a desired pose, an RGB-D image is rendered from the NeRF model. The RGB, depth, and reference sensor background images are then encoded through separate encoders. The outputs of these encoders are concatenated before being fed to the generator network to generate the desired tactile image. During training, a classifier is used to differentiate whether a tactile image is generated by the generator (fake) or is a ground-truth tactile image (real). The classifier loss and an L1 image loss are used to train the generator during training.

## 3.2 Conditional generative adversarial network

After using the NeRF model to render the RGB-D image of an object from a target camera pose $(\mathbf{o}, \mathbf{d}_c)$, the goal is to generate the corresponding tactile image. We define the matching criteria to be where the tactile sensor orientation is the same as the camera orientation, and the tactile sensor position is at a point along the centre camera ray $(\mathbf{s}, \mathbf{d}_c)$, where $\mathbf{s} = \mathbf{o} + t_s \mathbf{d}_c$ and $t_s$ is dependent on the geometry of the object.

We employ a conditional Generative Adversarial Network to learn the mapping $G : \{\mathbf{x}, \mathbf{z}\} \rightarrow \mathbf{y}$ between a rendered RGB-D image $\mathbf{x}$ and noise $\mathbf{z}$, on one hand, and the corresponding tactile image $\mathbf{y}$, on the other [27]. Taking inspiration from prior work [10], we modify the cGAN model to condition the generative model on the RGB image, the depth image, and a reference background image $\mathbf{b}$ using separate encoders, with skip-connections added between the encoder for the reference background image and the generator (see Figure 2 for detailed architecture). Following the cGAN literature [27], the noise $\mathbf{z}$ can be implemented using Dropout [28] in the generator network $G$. The objective function of the cGAN model is

$$G^* = \arg \min_G \max_D \mathcal{L}_{\text{cGAN}}(G, D) + \lambda \mathcal{L}_{\text{L1}}(G) \tag{2}$$

where

$$\mathcal{L}_{\text{cGAN}}(G, D) = \mathbb{E}_{\mathbf{x}, \mathbf{y} \sim p(\mathbf{x}, \mathbf{y})}[\log D(\mathbf{x}, \mathbf{y})] + \mathbb{E}_{\mathbf{x}, \mathbf{z} \sim p(\mathbf{x}, \mathbf{z})}[\log(1 - D(\mathbf{x}, G(\mathbf{x}, \mathbf{z})))] \tag{3}$$

and

$$\mathcal{L}_{\text{L1}}(G) = \mathbb{E}_{\mathbf{x}, \mathbf{y}, \mathbf{z} \sim p(\mathbf{x}, \mathbf{y}, \mathbf{z})}[\|\mathbf{y} - G(\mathbf{x}, \mathbf{z})\|_1] \tag{4}$$

## 4 Experiments

Experiments are first performed in simulation as proof-of-concept and then in the real world to test the hypothesis that generated tactile data can improve performance on downstream robotics tasks. To collect tactile images, we employ Digit tactile sensors [14]. These sensors have two main components: an RGB camera and a compliant layer made of a soft gel. As a working principle, the camera captures the deformations of the soft gel caused by contact forces. For the simulation experiments, we use TACTO to simulate the Digit tactile readings [15]. We note that the TACTO simulator uses PyBullet [29] as its physics engine and PyRender [30] as the rendering engine.

### 4.1 Dataset generation

*Simulation* For simulation experiments, we select 27 common household objects from the scanned YCB dataset [31]. To train a NeRF model for each object, we collect 48 images taken from evenly sampled poses around a sphere centred on the object. To perform a realistic collection of tactile readings, we randomly sample initial positions on a sphere centred on the object and move the tactile sensor from the initial position towards the centre of the object. To obtain tactile readings, a constant force is applied to the simulated Digit sensor, with the objects fixed. We discard cases when no reading is obtained, and collect 500 touches per object. Correspondingly, 500 RGB-D images are synthesised from the NeRF model for each object, using the sensor orientation as the view orientation. The rendered RGB-D images are then preprocessed and paired with their corresponding tactile images for training the cGAN model. For evaluation, we hold out 3 distinct objects as the novel object test set and use 24 objects for training. We also collect a validation set and a novel view test set of 50 paired tactile and RGB-D images from novel orientations for each training object.

*Real-world* For experiments in the real world, we select 9 common household objects from both the YCB dataset and other objects belonging to similar categories [31]. For training the NeRF models, we collect 118 images taken from camera poses that are uniformly sampled within a range in the hemisphere around the object using the set-up in Figure 1.

To automate data collection, we use a 7-DoF Franka Emika Panda arm with an Intel RealSense D415 camera for taking images, and select the camera pose range based on the workspace of the robot. For collecting the tactile reading, as seen in Figure 1, we attach a Digit sensor on the end effector flange of the Panda arm, and control the arm to move towards the centre of the object. For training, we perform 132 touches per object and collect 50 frames per touch. We also discard the cases when

no reading is obtained. We then synthesise and preprocess the corresponding RGB-D images using the NeRF model and pair them with the tactile images for training the cGAN model. For testing, we hold out 3 distinct objects as the novel object test set, and we hold out 12.5% of the training dataset as the novel view test set. It should be noted that the test set is randomly chosen using the index of the touch and not the frame. The resultant training dataset contains 398 touches and 19,900 frames. For each object, collecting images for the NeRF models takes approximately 20 minutes, whereas the collection of tactile data takes over 2 hours. The details of the objects are given in the Appendix.

All the objects in the simulated and real-world datasets are assumed to be rigid. For soft objects, their deformation when subjected to an external force, which is challenging to model, needs to be taken into account. By considering rigid objects only, the tactile sensor response mainly depends on the deformation of the soft layer of the sensor and on the contact force applied.

### 4.2 Evaluation details

As losses in cGAN are not directly indicative of the quality of the training results, it is difficult to use them to evaluate the realism of the generated tactile images. It is also difficult to manually evaluate the realism of such tactile images. Thus, we employ a set of approximate metrics and an example task to evaluate the quality of the generated images. We first evaluate the Structural Similarity Index (SSIM) and the Mean Squared Error (MSE) between the generated image and the ground truth tactile image on the hold-out test sets. Note that the average SSIM across RGB channels is taken, and that in simulation, ground-truth images refer to the simulated tactile images from the TACTO simulator. The SSIM is a reference metric with a range of $[0, 1]$, whereas the MSE measures the difference in pixel values with a range of $[0, 65025]$, since we consider 8-bit images [32]. Next, we use tactile classification as an example task to evaluate the usefulness of the generated tactile datasets. The same evaluation pipelines are applied to both simulation and real-world experiments. In evaluation, we scale all images to $128 \times 128$ which is also the size of the output of the cGAN model.

For both simulation and real-world experiments, we evaluate the generated images in a tactile classification task, using a simple convolutional neural network to perform object classification based on one tactile image. For comparison, we prepare two training datasets, one consisting of only ground-truth tactile images, the other consisting of both ground-truth and generated images. The test dataset consists of only ground-truth images. This is performed for both simulation and real-world data.

We also implement two benchmarks for comparison: 1) conditional GAN network for tactile image generation proposed by Lee et al. [9] 2) CycleGAN network [33]. In our experiments, both networks take RGB images as inputs and output tactile images at the corresponding pose. We use the default configurations proposed by the authors for our experiments.

## 5 Results and Discussion

We evaluate the results of tactile image generation in both a simulated setting and a real-world setting through a set of quantitative metrics and a tactile classification task. We also evaluate the results of transferring the learned cGAN model to a different sensor.

### 5.1 Simulation results

From Table 1, we can see that our approach is able to achieve high SSIM and low MSE values, indicating that the generated tactile images are structurally similar to the ground-truth (simulated) tactile images for both novel views of training objects and novel objects. This is corroborated by the qualitative results shown in Figure 3. On the novel view dataset, our approach is able to generate tactile images very similar to ground-truth ones, without having to explicitly model the object or the tactile sensor. The higher MSE value for novel objects could be due to a difference in the position of the predicted indentation, which leads to more differences with the ground-truth pixel-wise, as seen from Example 1 and 3 difference images in Figure 3, which indicate position mismatch.

To evaluate the quality of the generated tactile images, we use tactile object classification as an example task. By augmenting a given tactile dataset with generated tactile images, we are able to improve the success rate of the tactile classifier by a large margin. This shows that the proposed generative model is able to generate tactile images useful for the tactile classification task. Through

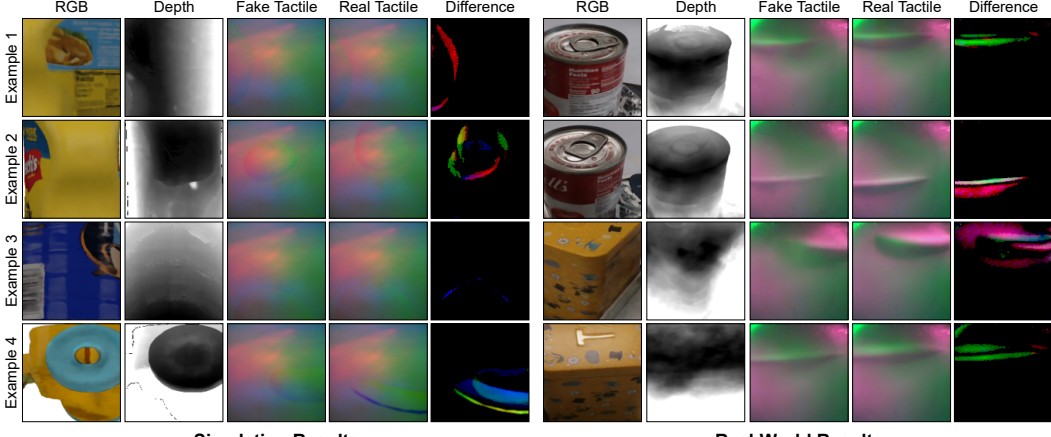

| | RGB | Depth | Fake Tactile | Real Tactile | Difference | RGB | Depth | Fake Tactile | Real Tactile | Difference |
|---|---|---|---|---|---|---|---|---|---|---|

**Simulation Results**        **Real-World Results**

Figure 3: Qualitative results on novel object test sets. Our approach generates faithful tactile images in both simulated (left) and real-world (right) domains on objects it has not seen before in training. The columns show RGB images, depth images, fake tactile images, real tactile images, and the difference images between the real and the fake images. The difference images are thresholded to remove noise. The differences are mostly due to a mismatch in position or amount of indentation and exist mostly around the edges of the indentation. The fake tactile images are model generated; the real tactile images provide ground truth.

| Test set | Method | SSIM ↑ | MSE↓ |
|---|---|---|---|
| Novel view | T-N | $0.980 \pm 0.001$ | $14.5 \pm 0.8$ |
| | Lee | $0.678 \pm 0.002$ | $1040 \pm 10$ |
| | Cycle | $0.649 \pm 0.002$ | $1660 \pm 10$ |
| Novel object | T-N | $0.961 \pm 0.000$ | $84 \pm 3$ |
| | Lee | $0.687 \pm 0.001$ | $990 \pm 10$ |
| | Cycle | $0.660 \pm 0.002$ | $1610 \pm 10$ |
| BG | - | $0.921 \pm 0.000$ | $67.4 \pm 3$ |

(a) SSIM values between generated tactile images and ground-truth (simulated) tactile images.

| Dataset | Accuracy/% ↑ |
|---|---|
| Sim | $85 \pm 3$ |
| Sim + T-N | $96 \pm 2$ |
| Sim + Lee | $86 \pm 4$ |
| Sim + Cycle | $80 \pm 0$ |

(b) Classification results for simulated dataset.

Table 1: Evaluation results for generated tactile images in simulation. Results shown are average values with standard errors. T-N: Touch NeRF (Ours). BG: Reference sensor background image.

leveraging NeRF models, we are also able to render additional RGB-D images of the object from novel view angles and generate additional tactile images to further augment the dataset at a low cost. Compared to the benchmarks, our approach outperforms by a large margin in both quantitative metrics and the classification task. This is likely due to the conditioning on depth, which enables our approach to predict tactile responses on 3D objects.

### 5.1.1 Transferring to a different sensor

By using a much smaller fine-tuning dataset than the original RGB-D and tactile training dataset, we are able to fine-tune a trained cGAN model to adapt to a different sensor with different characteristics. To demonstrate this, we select the OmniTact sensor as an example target tactile sensor for fine-tuning [19]. We collect simulated OmniTact images using a similar procedure as described in Section 4.1 and pair the tactile image with the corresponding RGB-D image rendered from NeRF models. We collect 5 tactile images for each training object for training and 50 each as the novel view test set. We hold out the same 3 objects as the novel object test set. We then continue to train the cGAN model that has been pre-trained on the simulated Digit dataset with the new OmniTact dataset, and evaluate the results using similar metrics as described in Section 4.2.

Using a much smaller fine-tuning dataset, the generated OmniTact tactile images still resemble the ground-truth (simulated) images, as shown in the SSIM and MSE results in Table 2a, and qualita-

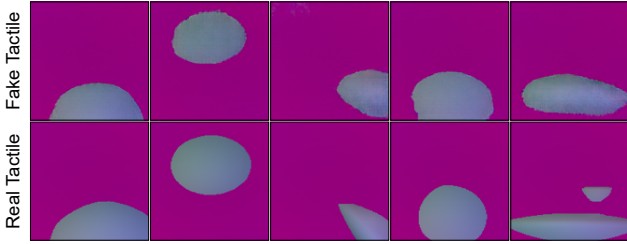

Figure 4: Example of generated tactile images for the simulated OmniTact sensor. In this case, the real tactile images are generated by the simulator.

| Test set | Method | SSIM ↑ | MSE↓ |
|---|---|---|---|
| Novel view | T-N | $0.865 \pm 0.002$ | $550 \pm 10$ |
| | FS | $0.829 \pm 0.002$ | $950 \pm 20$ |
| Novel object | T-N | $0.839 \pm 0.003$ | $760 \pm 30$ |
| | T-N+ | $0.847 \pm 0.003$ | $760 \pm 30$ |
| | FS | $0.811 \pm 0.005$ | $1300 \pm 60$ |
| BG | - | $0.856 \pm 0.001$ | $1390 \pm 20$ |

(a) SSIM values between generated tactile images and ground-truth (simulated) tactile images.

| Dataset | Accuracy/% ↑ |
|---|---|
| Sim | $86 \pm 0$ |
| Sim + T-N | $94 \pm 1$ |
| Sim + FS | $82 \pm 1$ |

(b) Classification results for using fine-tuning dataset.

Table 2: Evaluation results for generated tactile images for a new sensor in simulation. Results shown are average values with standard errors. T-N: Touch NeRF (Ours) with fine-tuning. T-N+: Touch NeRF with additional fine-tuning data. BG: Reference sensor background image. FS: Touch NeRF model trained from scratch.

tively in Figure 4. The lower SSIM value for novel objects is likely due to the presence of artefacts in the generated images. This could be remedied by adding an additional amount of fine-tuning data, as seen in the T-N+ results in Table 2a, which include an additional 45 training images (∼1/3 of the size of the original fine-tuning dataset). From Figure 4, it can also be seen that the generated tactile images are able to capture the shape of the indentation. Additionally, the background of the tactile image is accurately replicated, pointing to the benefit of conditioning the generative model on a reference sensor background image. Classification results in Table 2b demonstrate that the generated tactile images for the new sensor are still useful for increasing the accuracy of the tactile classifier by augmenting the tactile dataset. The fine-tuned cGAN model also outperforms the benchmark cGAN model trained from scratch, demonstrating the usefulness of pre-training and pointing to the potential of the proposed approach in leveraging available tactile datasets for different sensors to improve performance. This offers a potential avenue for further improving the data efficiency in tactile data generation using our approach.

## 5.2 Real-world results

From Figure 3 it can be seen that the proposed framework is able to generate realistic tactile images in the real world on both novel views of training objects as well as objects that it has not seen before during training. It is also able to generate the change in lighting caused by the object pressing against the soft gel. The difference in the amount of indentation between the generated image and the ground-truth image could be due to the variation of the force applied, as we only require the force to be within a range during tactile data collection.

Table 3 shows that the generated tactile images are able to capture the key properties of ground-truth tactile images collected in the real world. The SSIM values are lower than those in simulation in Table 1, which is expected due to the presence of noise in the real world. Additionally, classification results in Table 3b indicate that the generated tactile images are useful in increasing the accuracy of the downstream tactile classification task. The failure modes exhibited in the classification experiments are found to be mostly due to the similar tactile features exhibited in some test objects.

It can be seen that our approach outperforms both benchmarks in quantitative metrics and the classification task. It should be noted that the RGB-D images rendered by real-world NeRF models

contain more artefacts than those in simulation, as seen in Figure 3. However, even using the noisier RGB-D renderings, our approach is still able to generate high-quality tactile images for 3D objects. Additionally, our approach offers more generalisability on two fronts: firstly, as seen from the results in Table 3a, our approach outperforms in the novel object test set by a large margin; secondly, by leveraging a NeRF model, our approach is able to generate high-quality tactile images from novel viewpoints. This further removes the need to manually retake images for every new tactile image, as required by the baseline approaches.

| Test set | Method | SSIM ↑ | MSE↓ |
|---|---|---|---|
| Novel view | T-N | $0.887 \pm 0.001$ | $56 \pm 1$ |
|  | Lee | $0.708 \pm 0.006$ | $960 \pm 50$ |
|  | Cycle | $0.781 \pm 0.006$ | $420 \pm 60$ |
| Novel object | T-N | $0.838 \pm 0.001$ | $225 \pm 3$ |
|  | Lee | $0.723 \pm 0.003$ | $850 \pm 30$ |
|  | Cycle | $0.765 \pm 0.002$ | $390 \pm 10$ |
| BG | - | $0.814 \pm 0.000$ | $239 \pm 3$ |

(a) SSIM values between generated tactile images and ground-truth (real) tactile images.

| Dataset | Accuracy/% ↑ |
|---|---|
| Real | $74 \pm 1$ |
| Real + T-N | $83 \pm 2$ |
| Real + Lee | $76 \pm 2$ |
| Real+ Cycle | $70 \pm 0$ |
| T-N* | $70 \pm 6$ |

(b) Classification results for real-world dataset.

Table 3: Evaluation results for generated tactile images in the real world. Results shown are average values with standard errors. T-N: Touch NeRF (Ours). BG: Reference sensor background image. T-N*: Touch NeRF deployed 'in the wild' (training dataset contains no real data).

To illustrate the proposed approach in deployment, we collect RGB image data (48 images per object) for three additional objects on a different background using a mobile phone and trained corresponding NeRF models (with a lower resolution setting for faster training). We then generate tactile images for each object using our trained real-world cGAN model, and evaluate the quality of the generated tactile images in a similar classification task. The classifier is trained only on generated data, and is tested on manually-collected real tactile data. The T-N* result in Table 3b shows that the proposed approach is still able to achieve high accuracy in this more challenging setting. In this case, the data collection took around 2 minutes per object with casually taken camera images, further demonstrating the ease of deployment of our approach and the potential for reducing human efforts in tactile data collection.

## 6   Conclusion and Limitations

In this paper, we present a novel framework for the generation of tactile data for camera-based tactile sensors. The framework leverages NeRF models to render RGB-D images of an object from desired poses, and passes the RGB-D images to a cGAN model to generate the desired tactile images. Compared to state-of-the-art, this approach allows for the generation of tactile images for 3D objects from arbitrary viewpoints, without the need for accurate sensor or object models. Results demonstrate that the generated tactile images are structurally similar to ground-truth images, and are useful in downstream robotics tasks such as tactile classification. We further demonstrate the potential of a novel capability to transfer from one tactile modality to another with a small fine-tuning dataset.

One potential limit of the approach is that a NeRF model needs to be trained for each object. However, it should be noted that the collection of image data for training NeRF models requires much less human effort than the collection of tactile data, as illustrated by our deployment experiment. This is also arguably simpler than building a scanned object model. Additionally, recent advances in accelerating NeRF training also significantly mitigate the computation time needed [34].

We also assumed to operate only on rigid objects, with a fixed force range and orientation (pitch and yaw) of the tactile sensor. Soft objects might require modelling the non-linear deformation of the object when in contact, which adds additional complexity in predicting the tactile response. This is currently out of the scope of the paper. It must also be noted that even under the rigid object assumption, generating tactile images for 3D objects is challenging and is still an open problem for the tactile-sensing community. The generalisation to different forces and orientations, and potentially soft objects, is left to future work.

## Acknowledgments

We thank Jun Yamada for help with the experimental set-up, and Jack Collins and Alexander Mitchell for suggestions on the paper draft, and Yizhe Wu and other members of the Applied Artificial Intelligence Lab and Soft Robotics Lab for helpful discussions. We thank the anonymous reviewers for their valuable feedback in revising the paper. This work was supported in part by the UKRI/EPSRC Programme under Grant EP/V000748/1, and by a CSC-PAG Oxford Scholarship. We would like to acknowledge the use of the University of Oxford Advanced Research Computing (ARC) facility in carrying out this work. http://dx.doi.org/10.5281/zenodo.22558. We would also like to acknowledge the use of the SCAN facility.

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
