# OpenReview forum: "Touching a NeRF: Leveraging Neural Radiance Fields for Tactile Sensory Data Generation"
_robot-learning.org/CoRL/2022/Conference — CoRL 2022 Poster_

### Official Review · Reviewer_Vqs5 · 2022-07-29

**Originality:** Very Good
**Technical Quality:** Very Good
**Clarity Of Presentation:** Very Good
**Impact:** 3

**Recommendation:**

Weak Accept: I recommend accepting the paper, but will not argue for my recommendation if the majority of other reviewers have a different opinion.

**Summary:**

This paper proposes a technique for synthesizing tactile images from RGB scans of objects which are subsequently converted to NeRFs. The authors argue that simulators for generating tactile images are cumbersome to use because they rely on accurate object and sensor models, and are often specific to a single sensor type. To remedy this, they propose a method which uses cGANs to generate tactile readings from (1) a baseline tactile reading, and (2) an RGBD rendering from a trained NeRF of an object. Their experiments show good generation quality of the tactile images, and experiments on object classification suggest that using additional cGAN-generated data improves performance by around 10% compared to just using raw data.

**Issues:**

The writing suggestions should be addressed during revisions, and I think adding an additional real test object and comparing sim2real results would be reasonable in the time frame available, though understand that showing real-world transfer to a new sensor may be infeasible.

**Quality Of The Limitations Section:**

Additional details required

**Reviewer Expertise:**

4: The reviewer is confident but not absolutely certain that the evaluation is correct

**Robotics Focus:**

Sufficient demonstration on hardware

**Strengths And Weaknesses:**

Strengths

The paper is well-written, clear, and thorough, and I think the strategy of data-driven tactile synthesis is an approach worthwhile sharing with the community as an alternative to purely simulated approaches. The experiments are well constructed to illustrate the points argued in the text, and fairly presented.

Suggestions
- Though authors report that NeRF training time is no longer prohibitive, which is true, I feel a central weakness of the approach is its time-consuming real data collection process, requiring human involvement to gather and position training objects and initiate the captures, followed by 100s of sampled touches. Reporting time to collect data would be helpful to include, as well as some avenues to speed the process and reduce human involvement in the future.
- I feel it should be made more clear in the results tables and experiment descriptions that there are a different number of test objects for sim classification and real classification (2 vs 3), or ideally an additional test object should be added to the real tactile classification experiment to allow more meaningful comparison between sim and real results. The discussion states that "the higher absolute accuracy [of physical results] compared to the simulated results is likely due to the smaller number of objects for the real-world dataset.", which I agree with, but think this could be easily remedied by adding an additional test object to the real-world experiments.
- I would appreciate more detail in the classification experiments; namely, what specific objects were used for sim and real experiments, and what are some of the failure modes exhibited in both sim and real classification?
- Showing transfer to a different sensor in the real would be much more compelling than the simulated omnitact sensor results, if the hardware is available. Even showing transfer between different DIGITS would be interesting to show ability to capture small variations in sensor construction.
- It would be interesting to investigate sim2real performance to test the importance of real data. I suggest the following 2 comparisons to add to table 3, right-hand side: (1) sim data tested on real object test set, (2) sim+gen data tested on real object test set. Although real data is presumably more accurate, the simulated set has more data (500 touches per objects vs 132, 24 objects vs 6), so I would be curious to see its performance in the real world.

**Summary Of Recommendation:**

I recommend accepting the paper, and make some suggestions to improve clarity and enhance the experiments in the previous section.

---

> ### Author Response · Authors · 2022-08-25
> **Response to Reviewer Vqs5**
>
> We thank the reviewer for the constructive feedback, including the very positive assessment overall regarding our problem setting, approach, evaluation and presentation. The principal queries raised relate to the efficiency of the data collection process, the number of objects used for evaluation in real and simulation settings, details of our classification approach, transfer between real-world sensors and sim2real performance on real data. Below we address each of these in turn.
>
> **Time-consuming real data collection process**
>
> To quantify the advantage of our approach over manual image collection, we now include time for data collection in Section 4.1 [line 169-171], and in Section 5.2 for our additional deployment experiment [line 282-285]. To summarise, in training, for each object, collecting images for the NeRF models takes approximately 20 minutes, whereas the collection of tactile data takes over 2 hours. In deployment, our approach requires as little as 2 minutes for data collection for a new object.
>
> A strength of our approach is that tactile data collection and cGAN training need to be done only once. As we highlight in the conclusions section [line 296-300], for each novel object we only need to collect RGB images and train NeRF models (which has recently become much faster). We further illustrate the ease of this with the deployment experiment in which we acquire images of objects ‘in the wild’ using a mobile phone, train NeRF models on the phone-captured images, and generate novel tactile images for tactile classification. This reduces the human time required for data collection to 2 minutes per new object. We hope this corroborates the practical utility of our approach, reducing the human effort required for tactile data collection.
>
> **Different number of objects for sim and real classification**
>
> We thank the reviewer for this suggestion. As suggested, we have added an additional real test object, so there are now 3 simulated test objects and 3 real test objects in the classification experiments (3 vs 3).
>
> **Classification details**
>
> For reasons of space and flow of the narrative, these details are provided in the Appendix. To address the reviewer’s concern, we added an additional reference to the Appendix material in the main text [line 171], with added explanation of the failure modes exhibited in the classification experiments as suggested. [line 265-266] In brief, failure modes were mostly due to similar tactile features exhibited by some test objects (e.g. the edges of the tea box and edges of the soup can cause a similar tactile signature).
>
> **Transfer between real-world sensors**
>
> We thank the reviewer for this suggestion. Unfortunately, the hardware required for this is not currently available to us, however we do agree that this is an important step for future work.
>
> **Sim2real performance**
>
> We have experimented using a generated sim2real dataset (input: real-world NeRF-rendered RGB-D image; model: cGAN model trained in simulation) to augment the real-world dataset for training classifiers, and would be happy to report these results here. In quantitative metrics evaluated on real-world objects, we found that for test objects, SSIM = 0.710 $\pm$ 0.003, MSE = 1390 $\pm$ 20, whereas for train objects, SSIM = 0.726 $\pm$ 0.003, MSE = 1330 $\pm$ 30. However, it should be noted that the cGAN model is trained using only simulation objects, and thus has not seen the real-world train objects in training either. In the classification task, when using this new generated dataset for augmentation (i.e. real + sim2real gen), we found the accuracy to be 75 $\pm$ 2, representing a potential increase of 1% on the real-only dataset. However, these results are not currently included in the revised text because the performance largely depends on the quality of the tactile simulator that we use to collect data, and not on our approach, which renders them somewhat tangential to the main thrust of our submission.
>
> We also note that in order to apply tactile classifiers for sim2real transfer, the training dataset for the classifier (sim or sim+gen) needs to contain the same classes as the test dataset (real). However, in our experiments, the real-world object dataset and the sim object dataset contain different objects. Thus the sim2real evaluation in classification is currently infeasible.

---

### Official Review · Reviewer_gjCP · 2022-07-31

**Originality:** Poor
**Technical Quality:** Fair
**Clarity Of Presentation:** Very Good
**Impact:** 3

**Recommendation:**

Strong Reject: I recommend rejecting the paper and will argue for my recommendation even if other reviewers hold a different opinion.

**Summary:**

The paper proposes to train a generative 3D visual model (NeRF) of objects that it then queries at specific viewing angles to produce inputs to a conditional GAN with the intent to produce tactile signatures as would be experienced by applying a GelSlim sensor from that selected viewing angle. The premise is that this process would be effective in generating tactile data that would otherwise be expensive to collect on a real-world robotic system.

**Issues:**

Please see the weaknesses section, issues are clearly discussed there.

**Quality Of The Limitations Section:**

Limitations are not well addressed

**Reviewer Expertise:**

5: The reviewer is absolutely certain that the evaluation is correct and very familiar with the relevant literature

**Robotics Focus:**

Relevant but unlikely to deploy to hardware in near future

**Strengths And Weaknesses:**

Merits and Strengths:

The paper does indeed address an important challenge in manipulation: how do we generate tactile data for downstream robot learning tasks, given that it is so expensive to collect data? There are a lot of existing learning approaches that we can draw from given that the GelSlim’s and similar vision-based tactile sensors produce images and this paper recognizes that effectively. The paper does a good job of evaluating the proposed approach on a real-world robot experiment implemented with a Franka Emika Panda.

Weaknesses and Limitations:

The biggest concern with the paper is: What is the core technical contribution/novelty? The paper uses existing and well-established learning techniques (NeRFs + Conditional GANs) to generate realistic image-based tactile signatures. There was no need to adapt the NeRF or cGAN architectures as they both directly operate on images (and the conditioning vector can be any vector in R^n). The losses are standard, as is the training procedure and the data generation. The application (generating tactile signatures within distribution of real-world signatures) is interesting. However, without a technical nugget (e.g., novel learning/inference algorithm, novel architecture, novel simulator, novel functionality) there is not a sufficient advancement in our knowledge or state-of-the-art in robot learning to warrant publication in the manuscript's current state. To illustrate this point further, the methods section (where the core novelty of any paper should shine) is simply an overview of NeRFs and cGANs. Assuming the author is familiar with both, the entire section can be boiled down to 1 paragraph, suggesting only a very incremental contribution.

Beyond the above concern, the performance evaluation and benchmarking are lacking. It is true that evaluating the quality of tactile signatures is difficult. This is because there is a strong task dependence. Some tasks require ultra-fine detail while others only need very coarse detail. As such, it is difficult to draw any conclusions from the metrics used to evaluate the tactile signatures. The units for both metrics are missing and there is no effort in contextualizing beyond the idea that small numbers are better. We get no intuitive feel for how much more of an improvement the numbers suggest. The improvement in the downstream task is also relatively modest. Finally, there are many generative approaches the paper could benchmark against and those would have made fine benchmarks but there is no attempt in the manuscript.

The success rate in the Franka Kitchen environment is low for all models, and there is a modest (about 15% based on the graph) improvement but nothing to suggest that R3M is the solution. The Meta Worlds results are inconclusive, the performances are quite similar. Adroit is between the two, R3M does better but it's within 10% of the next baseline. We cannot conclude that R3M is a significant advance on the state-of-the-art based on these results.

The limitations section is, by far, the worst I’ve read in the several papers I have reviewed for CoRL and is certainly not up to par with the intent of the prompt provided by the conference organizers. This section is 6 sentences of which the first is a paper summary and the second is a filler. The second stated limitation is particularly sticking out. The purpose of the paper is to mitigate the requirements for data collection since it is expensive; however, the model needs to be retrained per new object. This is an important limitation that requires much more discussion than a throw away citation. I don’t really see the conclusion section adding any value to the paper, there is no new information presented. One approach could be to remove it and have a proper discussion and limitations section. Finally, I would want to see a clear statement on the biggest assumption: rigidity of the object. The tactile signature is only really predictable from the image because the object does not comply. This should receive far more attention and be clearly stated + implications noted.

General Comments:

The manuscript provides some encouraging preliminary results; however, without a core novel contribution (whether it is algorithmic, architecture, simulation/modeling or something to this effect) it is not ready for publication.


**Summary Of Recommendation:**

The manuscript provides some encouraging preliminary results; however, without a core novel contribution (whether it is algorithmic, architecture, simulation/modeling or something to this effect) it is not ready for publication. Simply put, concatenating two existing approaches with essentially no modification for the purpose of an application is not a sufficient contribution.

---

> ### Author Response · Authors · 2022-08-25
> **Response to Reviewer gjCP (2/2)**
>
> **Downstream improvement and benchmarks**
>
> We thank the reviewer for the suggestion on adding benchmarks. We have added two additional baselines: one conditional GAN approach from the tactile literature [1], and a new CycleGAN baseline [2]. Our approach is able to outperform both benchmarks by a large margin in both quantitative metrics and the tactile classification task. [lines 226-229,267-275] The poor results of the benchmarks shown in Table 1 and 3 illustrate the challenge of the task. Thus, even though the improvement in accuracy might appear modest in numbers, we believe this still represents a significant advance in the field.
>
> **Robotics focus: unlikely to deploy to hardware in near future**
>
> We would like to highlight that we have demonstrated our approach in real-world with a Panda arm and a real Digit tactile sensor, as noted by the reviewer. To further demonstrate that the method can be deployed quite easily, we added a new deployment experiment to the paper showing how we use only a smartphone camera and generate novel tactile images. [line 276-285]
>
> **Limitations**
>
> We thank the reviewer for the comments and suggestions on the limitation section. Following the recommendations, we have revised the limitations section and merged it with the conclusions. [line 287-307]
>
> **Model re-training**
>
> We would like to highlight that the cGAN model only needs to be trained once, and that even though a NeRF model needs to be trained for a new object, data collection for NeRF is much easier compared to tactile data collection. We further demonstrate this with our deployment experiment [line 276-285]. It should also be noted that previous approaches require a new image to be manually taken for each tactile image, even in deployment. We remove this need through the use of NeRF models.
>
> **Rigid object assumption**
>
> We thank the reviewer for pointing out this limitation. We have added a discussion of this in Section 4.1 [line 172-175], and in Section 6 [line 301-304]. We would like to additionally remark that even with rigid objects, the tactile signature is challenging to predict. It depends on a number of factors, such as the force applied on the object which changes the level of details captured by the sensor. Additionally, the soft gel embedded in the Digit sensor acts as a mechanical low-pass filter, and thus the tactile output is also affected by the mechanical properties of the sensor itself. This is also added in the manuscript. [line 137-139] Hence, even rigid objects already pose complex challenges and generating tactile data in these conditions is still an open challenge. Compared to the benchmarks, our proposed approach has achieved a significant improvement in generation results.
>
> **Franka Kitchen**
>
> We think this comment is not related to our paper.
>
> [1] Lee et al. “Touching to See” and “Seeing to Feel”: Robotic Cross-modal Sensory Data Generation for Visual-Tactile Perception. ICRA 2019.
>
> [2] Zhu et al. Unpaired Image-to-Image Translation using Cycle-Consistent Adversarial Networks. ICCV 2017.
>
> [3] Si et al. Taxim: An Example-Based Simulation Model for GelSight Tactile Sensors. RA-L 2022.
>
> [4] Gomes et al. Generation of GelSight Tactile Images for Sim2Real Learning. RA-L 2021.
>
> [5] Jianu et al. Reducing Tactile Sim2Real Domain Gaps via Deep Texture Generation Networks. ICRA 2022.
>
> [6] Ding et al. Sim-to-Real Transfer for Optical Tactile Sensing. ICRA 2020.
>
> [7] Agarwal et al. Simulation of Vision-based Tactile Sensors using Physics based Rendering. ICRA 2021.
>
> [8] Gao et al. ObjectFolder 2.0: A Multisensory Object Dataset for Sim2Real Transfer. CVPR 2022.

---

> ### Author Response · Authors · 2022-08-25
> **Response to Reviewer gjCP (1/2)**
>
> We thank the reviewer for the constructive feedback, including the positive assessment regarding our motivation and real-world experimental evaluation. The principal queries raised relate to the technical contribution, the baselines, and the limitations. Below we address each of these in turn.
>
> **Technical contribution: lack of novel functionality**
>
> The contribution of our submission is a novel approach to the generation of tactile images on 3D objects. We demonstrate the efficacy of our approach by evaluating the results using quantitative metrics and a downstream tactile classification task, and by comparing our results with newly added baselines. Our method outperforms the state-of-the-art in tactile data generation by a large margin. These gains are realised by applying learning methodology to a novel and complex problem in robotics leading to significant and important novel functionality. This is echoed by the other reviewers who highlight that the approach is ‘well-motivated’, ‘addresses an important and challenging problem in a practical way, offering a clever solution’, and ‘worthwhile sharing with the community’.
>
> Regarding architecture contributions, we have highlighted the adaptation of our architecture [line 126-129]. Concretely, we modified a standard cGAN architecture to condition on RGB and depth and background image.
>
> More importantly, our approach demonstrates two important novel functionalities. Firstly, we are one of the first to generate tactile images from RGB-D data on 3D objects. The poor results of the benchmarks in Tables 1 and 3 illustrate the significance of the challenge involved, because the generation of tactile images in 3D is not a classic direct image-to-image translation problem. By combining two methods in a novel framework, our approach is able to deal with this challenge in a simple yet effective way.
>
> Additionally, we would like to highlight that our paper is also the first to demonstrate transfer across different sensors, which has been a challenge in the tactile community owing to the different modalities involved. This novel capability opens up the potential for consolidating and leveraging existing tactile datasets, and has the potential to advance the use of learning-based techniques in the tactile domain.
>
> We believe that these are meaningful advances in an area of robotics that is underserved. We leave it up to the reviewer/AC discussion to determine what consensus exists that novelty in robot learning is really limited to novel architectures or losses.
>
> **Details and evaluation metrics**
>
> We agree that the amount of detail required depends on the task at hand. However, our evaluation demonstrates that our approach generates information appropriate for a downstream classification task as indicated by common metrics in the literature. We would therefore argue that this provides a successful proof of concept.
>
> **Units for metrics and context**
>
> We thank the reviewer for the suggestion, and we included an explanation of the SSIM and MSE metrics in Section 4.3 [line 191-193]. We note that these two metrics are commonly used in the literature [1,3,4,5,6,7,8], and we thus kept them to facilitate comparisons. We hope that the classification example, the comparison against the benchmarks, and the additional difference image in Figure 3 would be useful for contextualising the results. We have added the unit for the accuracy measure, and we also note that the SSIM values are reference metrics without units, and the MSE values represent pixel values squared.

---

### Official Review · Reviewer_ozxa · 2022-08-09

**Originality:** Very Good
**Technical Quality:** Very Good
**Clarity Of Presentation:** Very Good
**Impact:** 4

**Recommendation:**

Strong Accept: I recommend accepting the paper and will argue for my recommendation even if other reviewers hold a different opinion.

**Summary:**

This paper proposes a framework for generating synthetic tactile data, since collecting data on a physical system is tedious and simulators lack of accurate models. The method leverages a trained NeRF model to generate novel views of a scene, and a cGAN to generate the tactile data given a rendered NeRF view. To evaluate the quality of the synthetic tactile images they use SSIM and MSE, and use the generated data to train a classifier on a physical robot.

**Issues:**

The weaknesses.

**Quality Of The Limitations Section:**

Limitations are addressed clearly

**Reviewer Expertise:**

4: The reviewer is confident but not absolutely certain that the evaluation is correct

**Robotics Focus:**

Sufficient demonstration on hardware

**Strengths And Weaknesses:**

Strengths:
1) The paper is clear, easy to follow and the problem is well motivated.

2) This paper addresses an important and challenging problem in a practical way, offering a clever solution. It proposes a way to scale data collection for tactile sensors without modelling the sensor directly, potentially saving time and effort compared to physical data collection.

3) The paper demonstrates generalization properties by transferring learning from a specific sensor to a different one through fine-tuning, which is important since it shows we can leverage tactile datasets for different sensors.

Weaknesses:
1) In figure 3 it’s hard to understand the difference between the synthetic and real tactile images, perhaps the authors can add a disparity/difference image between the two to emphasize it.

2) In 5.1.1 the authors mention that the models get lower SSIM on novel objects due to artifacts in the synthetic images that can disappear with additional fine-tuning data. If these results exist, then adding them to the table would strengthen this argument.

3) The results in tables 1 and 2 could be made clearer by computing normalized/intuitive metrics (% for example).


**Summary Of Recommendation:**

I recommend accepting the paper. Not only that it addresses an important problem, but it also presents and interesting use of NeRF for tactile data generation.

---

> ### Author Response · Authors · 2022-08-25
> **Response to Reviewer ozxa**
>
> We thank the reviewer for the kind, thoughtful, and constructive feedback. We have implemented the following changes as suggested (1) include difference images in Figure 3 to emphasise the difference between synthetic and real tactile images; [line 194] (2) show improvement in transfer results with a larger (~⅓ larger) fine-tuning dataset in Table 2; [line 242-245] and (3) included an explanation of the SSIM and MSE metrics in Section 4.3 [line 191-193]. We note that these two metrics are commonly used in the literature [1,2,3,4,5,6,7], and we thus kept them to facilitate comparisons.
>
>
>
> [1] Lee et al. “Touching to See” and “Seeing to Feel”: Robotic Cross-modal Sensory Data Generation for Visual-Tactile Perception. ICRA 2019.
>
> [2] Si et al. Taxim: An Example-Based Simulation Model for GelSight Tactile Sensors. RA-L 2022.
>
> [3] Gomes et al. Generation of GelSight Tactile Images for Sim2Real Learning. RA-L 202.
>
> [4] Jianu et al. Reducing Tactile Sim2Real Domain Gaps via Deep Texture Generation Networks. ICRA 2022.
>
> [5] Ding et al. Sim-to-Real Transfer for Optical Tactile Sensing. ICRA 2020.
>
> [6] Agarwal et al. Simulation of Vision-based Tactile Sensors using Physics based Rendering. ICRA 2021.
>
> [7] Gao et al. ObjectFolder 2.0: A Multisensory Object Dataset for Sim2Real Transfer. CVPR 2022.

---

### Official Review · Reviewer_Fy6z · 2022-08-10

**Originality:** Good
**Technical Quality:** Very Good
**Clarity Of Presentation:** Excellent
**Impact:** 3

**Recommendation:**

Weak Accept: I recommend accepting the paper, but will not argue for my recommendation if the majority of other reviewers have a different opinion.

**Summary:**

This paper presents a method for generating tactile data for objects using a conditional GAN, with training data coming from NeRF trained on  comparatively small amounts of RGB-D data but separately for each object. The motivation comes from the fact that tactile data from sensors is hard to collect and generalize across hardwares. Related approaches use GANs directly from visual data, but this requires a lot of images.

Approach: NeRF is trained separately for each object, then used to generate various RGB-D images. The cGAN is trained on a set of NeRF-generated RGB-D + ground truth tactile images, then is used to generate tactile images from the NeRF-generated RGB-D (in turn generated from a desired pose) and a predetermined background image.

On holdout object experiments, the proposed approach reaches seemingly high SSIM wrt ground truth. MSE is more varied, but low for novel views at least. Using this method for fine-tuning shows some performance improvement as well.

**Issues:**

- Lack of baselines, comparisons, or analysis of value/potential impact


**Quality Of The Limitations Section:**

Limitations are addressed clearly

**Reviewer Expertise:**

3: The reviewer is fairly confident that the evaluation is correct

**Robotics Focus:**

Sufficient demonstration on hardware

**Strengths And Weaknesses:**

**Strengths**
- Paper is exceptionally clear and well-written. An interesting but easy read, which is ideal. Congrats to the authors on a very well-presented work!
- The approach is intuitive and has a clear value proposition. The advantages with respect to training data efficiency are evident, and the paper as a whole is well-motivated given both the baseline approach of actual tactile sensors as well as more related approaches using cGANs.
- While the experimental section isn't particularly thorough, it's well designed. It's good to see holdout experiments and comparison between novel view and novel object

**Weaknesses**
- There doesn't seem to be any baseline in this paper. I realize that the value isn't exactly degree of improvement on some metric, but this is ultimately a new approach to an existing problem that should make its value evident. This can include even a brief analysis on comparison of data efficiency, training time, and generality between many single-object NeRFs vs. the prior cGAN work. Even if performance from other methods is better, an argument can be made for why this one is worthwhile/worth iterating on
- Related to this, we are essentially viewing the results in a vacuum. This makes it difficult to judge whether a user should try this method
- Figure 3 is helpful, but ultimately readers without a lot of experience with tactile methods won't know whether the tactile comparison is particularly good or not. Perhaps this paper isn't for such readers, but the venue certainly is. The fake images look promising but with room for improvement compared to the real ones, but ultimately I don't know how to judge whether this is a step forward. It's even more unclear given that there may be increased data efficiency and increased compute requirements to consider.
- This has no bearing on my score, but I find the title somewhat off-putting even if it's a good pun.


**Summary Of Recommendation:**

I recommend weak accept because I don't see any major flaws in this paper and it seems useful, but I also am not convinced of its significance - not because I have a reason to believe it isn't significant, but because 1) the results aren't deeply surprising and 2) I have nothing to compare to. Someone with more domain knowledge will be able to provide a better assessment.

---

> ### Author Response · Authors · 2022-08-25
> **Response to Reviewer Fy6z**
>
> We thank the reviewer for the constructive feedback, including the very positive assessment overall regarding our presentation, motivation, and experiment designs. The principal queries raised relate to the baselines, the context, the figure design, and the significance of the results. Below we address each of these in turn.
>
> **Baselines and analysis**
>
> We have added two baselines which represent the state-of-art in tactile data generation: one from the tactile literature using a conditional GAN network [1], and one employing a CycleGAN [2]. [line 203-207]  In both cases we use the original authors’ own code and default configurations. Our approach outperforms both benchmarks in quantitative metrics, and in the tactile classification task by a significant margin. [lines 226-229, 267-275]
>
> We have also added additional discussion on data efficiency. We stress that our cGAN model only needs to be trained once. Further, for each object, once a NeRF model is trained, little additional human effort is required for the generation of tactile images. This is in contrast to previous approaches, which require the manual retake of camera images for every new tactile image. We have highlighted this in the manuscript. [lines 274-275] We also hope to add that the data collection and training for individual NeRFs in fact carries a very low overhead, as illustrated in the deployment experiment. [line 276-285]
>
> We further comment on the ability of our approach to generalise compared to the benchmarks, and added the following discussion [line 271-274] ‘our approach offers more generalisability in two fronts: Firstly, as seen from results in Table 3a, our approach outperforms in the novel object test set by a large margin. Secondly, by leveraging a NeRF model, our approach is able to generate high-quality tactile images from novel viewpoints.’
>
> **Contexts and benefits to users**
>
> We have added two baselines, and shown that our approach outperforms both by a large margin. We further demonstrated the ease of deployment of our proposed approach in an additional deployment experiment. [line 276-285] We hope these additions help to provide a context to our results in the challenging problem of tactile data generation for 3D objects, and to the practical application of our method.
>
> **Figure 3**
>
> As suggested by reviewer ozxa, we added difference images to Figure 3 for the real and simulated tactile images. [line 194] These should make the interpretation of Figure 3 more intuitive. We further highlight that these results outperform the benchmarks by a wide margin, noting that the benchmarks struggle in a 3D setting.
>
> **Importance and contribution**
>
> In response to the concerns that the reviewer has regarding the significance of our results, we would like to highlight two major advances that our paper achieves. Firstly, our paper is among the first learning-based approaches to generate tactile images for 3D objects empirically (without being given an object model). The fact that this is challenging can be seen from the benchmarks we now provide, which perform rather poorly. This is because the generation of tactile images in 3D does not conform to a classic direct image-to-image translation problem. The prediction of tactile response in 3D requires the model to, for example, identify the tactile features (e.g. edges) of the objects and ignore distractors such as colour, texture and backgrounds. In contrast to related work, this is specifically addressed in our approach by the conditioning on depth and sensor background. The significant margin between our results and the benchmarks emphasises the importance of our method to break a path forward for this challenging task.
>
> Secondly, our paper is also the first to demonstrate transfer across different sensors. This has exciting implications for consolidating and leveraging existing tactile datasets. Indeed, the different modalities and technologies involved in touch have limited the use of learning-based techniques in the tactile domain, as each research group needs to collect a new dataset for each sensor. Again, our proposed approach suggests a solution to this impasse. It is our hope that CoRL might provide a forum for us to show off the method’s capabilities to the community.
>
> [1] Lee et al. “Touching to See” and “Seeing to Feel”: Robotic Cross-modal Sensory Data Generation for Visual-Tactile Perception. ICRA 2019.
>
> [2] Zhu et al. Unpaired Image-to-Image Translation using Cycle-Consistent Adversarial Networks. ICCV 2017.

---

### Author Response · Authors · 2022-08-25
**Revised PDF for rebuttal**

We thank all the reviewers and the meta reviewer for the constructive comments. We have revised the paper to address the reviewers’ feedback. The major changes are highlighted in blue.

---

### Meta-Review · Area_Chair_XZtY · 2022-08-11

**Recommendation:** Accept (Poster)
**Confidence:** 4

**Metareview:**

The paper proposes a combination of NeRFs and cGANs as a means to generate synthetic but realistic tactile data, as would be seen from optical tactile sensors such as GelSight.  This is in order to save time on data collection for learning downstream tasks that operate on optical tactile data.

The reviewers agree that the paper is very clear and well-written, and that the method offers real value in easing a difficult data collection task in robotics. Most of the reviewers' original concerns have been well-addressed by the revised paper and the authors' rebuttal.  The paper will likely have value for the CoRL audience.

---

> ### Author Response · Authors · 2022-08-25
> **Response to Meta Reviewer**
>
> We thank the meta reviewer and the reviewers for the thoughtful and constructive feedback, for commenting on the quality of the writing, and for recognising the value of the robotics problem that we are trying to solve. The following major updates have been made to our submission. The corresponding major edits in the manuscript are also highlighted in blue.
>
> - As requested by reviewers Fy6z and gjCP, two baselines have been added to the paper: one is a conditional GAN approach from the tactile literature [1]; the other is CycleGAN baseline [2]. [line 203-207] Our approach outperforms the benchmarks in both quantitative metrics and the classification experiments by a significant margin. [lines 226-229,267-275]
>
> - As requested by reviewers Fy6z and ozxa, difference images have been added to Figure 3 [line 194]. An analysis is also added. [line 219-220]
>
> - As requested by reviewers ozxa and gjCP, Tables 1, 2, and 3 have been updated to indicate the desired direction of change to the reader. [lines 221,244,279] An additional explanation for the metrics is also added in Section 4.3. [line 191-193]
>
> - As requested by reviewer gjCP, the text has been revised to more clearly explain our architecture. [line 126-129]
>
> - As requested by reviewer gjCP, the limitation and conclusions sections have been expanded and merged. [line 287-307]
>
> - As requested by reviewers Fy6z, gjCP, and Vqs5, the text has been revised to quantify the human effort required for acquiring data with our method. [line169-171] A new experiment has also been added to demonstrate the ease with which our model can be deployed. Specifically, we show how tactile images can be generated for 3D objects ‘in the wild’, using only a smartphone camera for data collection. [line 276-285]
>
> - As requested by reviewer gjCP, an additional discussion on the rigid object assumption has been added. [line 172-175]
>
> - As requested by reviewer ozxa, additional results on the transfer experiments have been added. [line 242-245]
>
> - In response to suggestions from reviewer Vqs5, updates to the classification experiments have been implemented to: 1) add a real-world test object; [lines 156, 167, 403] 2) to more clearly separate the experiments into train and test objects; [line 407] and 3) to update the classification datasets to include an identical size for each object in both simulation and real-world scenarios to facilitate comparison. [line 413]
>
> - In response to suggestions from reviewers Fy6z and gjCP, the model results have been further improved with additional hyperparameter tuning.
>
> - The appendix is updated to reflect the above changes. The appendix is also attached to the main text for easier review.
>
> [1] Lee et al. “Touching to See” and “Seeing to Feel”: Robotic Cross-modal Sensory Data Generation for Visual-Tactile Perception. ICRA 2019.
>
> [2] Zhu et al. Unpaired Image-to-Image Translation using Cycle-Consistent Adversarial Networks. ICCV 2017.